# WIN: Variable-View Implicit LIDAR Upsampling Network

## Abstract

LiDAR upsampling aims to increase the resolution of sparse point sets obtained from low-cost sensors, providing better performance for various downstream tasks. Most existing methods transform LiDAR points into range view and design complex neighborhood point interpolation strategies to increase the resolution of point clouds. However, the range view presentation of LiDAR data provides only a single perspective, preventing a holistic understanding of the scene geometry. To address this issue, we propose WIN, a Variable-View Implicit lidar upsampling Network. First, we decouple the range view into two novel virtual view representations, Horizon Range View (HRV) and Vertical Range View (VRV), to compensate for the missing geometric information during interpolation. Secondly, our proposed two virtual views are orthogonal. The feature difference between the two views is proportional to the complementarity of the information between the two views. So, we introduce a Contrast Selection Module (CSM) that guides the selection process by capturing the feature differences between the different representations. In addition, we observe that it is difficult for CSM to predict the correct result because the label (which is the best view for each point) changes frequently during training. Therefore, we model the selection of the best views as a probability distribution problem. We predict the view confidence score rather than the categorization label. As a result, compared with the current state-of-the-art (SOTA) method ILN, WIN introduces only 0.4M additional parameters, yet achieves a +4.5% increase in the MAE on the CARLA dataset. Furthermore, our method also outperforms all existing methods in one downstream task (Depth Completion). The pre-trained model and code will be released upon acceptance.

## 1 Introduction

By capturing point clouds from the surrounding environment, Multi-beam LiDAR (MBL) plays an important role in various tasks, such as object detection Lang et al. (2019), mapping Chen et al. (2021b), and localization Wang et al. (2021). The density of the LiDAR point cloud directly determines the performance of these downstream tasks. However, the commonly used high-end MBL sensors with 64-128 scan lines have high power consumption and cost, limiting, limiting their wide applications Chen et al. (2024). As a result, LiDAR upsampling, serving as a low-cost and high-efficiency alternative, has attracted the attention of more and more researchers Tian et al. (2022); Savkin et al. (2022); Yang et al. (2024); Chen et al. (2022).

LiDAR Upsampling aims to increase the resolution of sparse point sets obtained by low-end MBL sensors (such as VLP-16). Different from some object-level point cloud upsampling methods Yu et al. (2018); Long et al. (2022); Lim et al. (2024), considering the characteristic line pattern of MBL points, the groundbreaking work projects LiDAR points into range view via spherical projection and obtains high-line LiDAR points by improving the image resolution Triess et al. (2019); Shan et al. (2020). After that, some researchers used powerful visual encoder backbones to directly model the mapping relationship from low-resolution range images to high-resolution range images, and achieved good performance Jung et al. (2022); Yue et al. (2021); Chen et al. (2021a); Yang et al. (2024). However, these explicit methods suffer from large parameters, low efficiency, and can only be used on fixed upscale factor, which limits their application scenarios. To this end, the implicit function methods set the task as a pixel interpolation mission, which upsample LiDAR points by searching neighbor points and predicting corresponding interpolation weights Kwon et al. (2022);

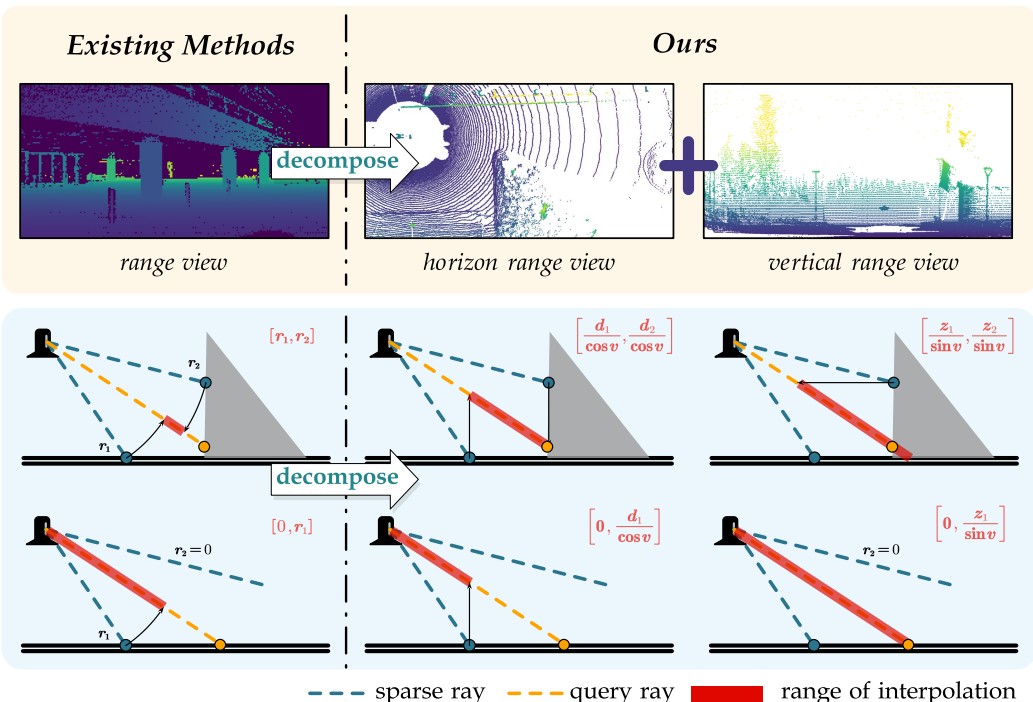

Figure 1: General overview of our approach. We decompose the range view into two orthogonal views (horizon range view and vertical range view), eliminating shape distortion in the distance view. At the same time, our method can benefit from both views, which avoid the limitations of a single view interpolation. At the bottom of the image, we plot two typical scanning scenarios (object edge and ground), as well as a schematic diagram of the interpolation capabilities in different views. Intervals that can be represented by interpolation are marked in red.

Park et al. (2023). These methods have greater flexibility and allow upsampling at any upsampling rate, thus gradually leading the development of this field.

However, these mainstream implicit methods do not recognize a fundamental question: *Is this range view representation suitable for interpolation-based LiDAR Upsampling?* As we know, the distributions of points are different in different views. While, the range view only reflects the distance between the observer and the points in the scene. This single geometric representation makes it difficult to accurately reflect the geometric character during interpolation, such as vertical or horizontal surfaces. As shown in Fig. 1, based on the existing implicit interpolation algorithm, we illustrate the possible interpolated position in red color. Due to the limitations of single-view representation, existing methods are insufficient to provide accurate interpolation results for critical boundary regions and non-smooth surfaces, thus causes the upsampled point cloud to contain shape distortions.

To address these problems, we propose a novel Variable-View Implicit LiDAR upsampling Network (WIN), which decouples the 3D representation of Range View (RV) into two novel virtual view representations Horizon Range View (HRV) and Vertical Range View (VRV), effective upsampling. Specifically, HRV is responsible for horizon range-based interpolation, which ignore the $z$ values, while VRV is cooresponding to vertical range-based, ignoring the $x$ and $y$ values. This key idea stems from the fact that HRV and VRV, as an orthogonal transformation of RV, can provide more perspectives for observation without losing any geometric information. Instead of designing complex feature fusion strategies for these views, we simply generalize the Implicit function methods Kwon et al. (2022); Park et al. (2023) to interpolate points in different views. It allows us to enjoy the advantages of variable-view representations without introducing unnecessary parameters or changing network architecture.

Furthermore, we observed that the interpolation points have strong geometric heterogeneity due to the orthogonal property between views. Specifically, horizon range $d$ interpolation performs better at

the vertical surface of the object, while in some other flat areas, vertical range $z$ has more advantages. This observation motivated us to design a contrast selection module, which help each interpolation point to choose the corresponding best view based on the geometric differences in the upsampled image from different representations.

However, during the training process, the classification label (which is the best view for each up-sample point) is constantly changing due to the different convergence speeds of different branches, making it difficult to supervise the selection module using a binary label. Therefore, we model the best view selection process as a probability distribution problem. Specifically, we set the result of the selection module as the confidence level of the view instead of the classification probability. We compute the distance between the predicted value and the ground truth, and define the truth confidence through a Gaussian distribution, thus achieving effective supervision for the selection module.

To sum up, we propose a novel LiDAR upsampling framework named WIN, illstured in Fig. 2. Extensive experiments shows that, WIN achieved SOTA performance on both virtual and real-world datasets. An improvement of 4.53% and 7.01% is achieved on MAE and IoU, respectively. For downstream task, we verified that the proposed method can significantly improve the accuracy of depth completion by low-resolution point clouds, and the improvement is greater than that of existing methods.

In general, our major contributions can be summed up as follows:

- We analyze the limitations of ranging image-based methods and design a lightweight novel network, WIN, for more efficient LiDAR Upsampling.

- We are the first to decouple the 3D representation of a ranging image into two orthogonal view representations. And, we propose a contrast selection module, which predicts a confidence score for each upsampled point, thus achieving effective LiDAR Upsampling.

- We evaluated our methods on large-scale synthetic and real-world datasets, it demonstrate that our WIN significantly improves interpolation accuracy while requiring minimal memory and computation time. Besides, our method also shows the best performance on one downstream task (Depth Completion).

## 2 RELATED WORKS

### 2.1 OBJECT-LEVEL POINT CLOUD UPSAMPLING

Point cloud upsampling is a fundamental task in 3D computer vision. Qi et al. proposed Point-Net Qi et al. (2017a) and PointNet++ Qi et al. (2017b), first using deep neural networks to handle disordered point cloud data. Afterward, PU-Net Yu et al. (2018) was proposed as the first learning method for point cloud upsampling. To improve the robustness, PU-Net splits the point cloud into patches and encrypts them through feature extraction, feature expansion, coordinate reconstruction, and patch merge. Subsequently, various upsampling algorithms were proposed to improve the performance on benchmark datasets Akhtar et al. (2022); Zhao et al. (2022); Feng et al. (2022); Qiu et al. (2022); He et al. (2023); Qu et al. (2024); Rong et al. (2024). However, these methods all focus on object-level point cloud data. When applied to large-scale LiDAR point clouds, they will bring huge computational burdens. Furthermore, the restricted receptive field limits the application of object-level solutions.

### 2.2 SCENE-LEVEL LIDAR UPSAMPLING

Considering the line-scanning characteristics of LiDAR point clouds, existing methods transform LiDAR points to the range image, and encrypt the point cloud through the image super-resolution. According to the consensus, we divide these LiDAR Upsampling methods into explicit-based and implicit-based methods.

**1) Explicit methods.** Explicit methods directly model the mapping relationship from low-resolution range images to high-resolution range images through the network. Thanks to the development of image super-resolution, some researcher tried to migrate image super-resolution methods to LiDAR

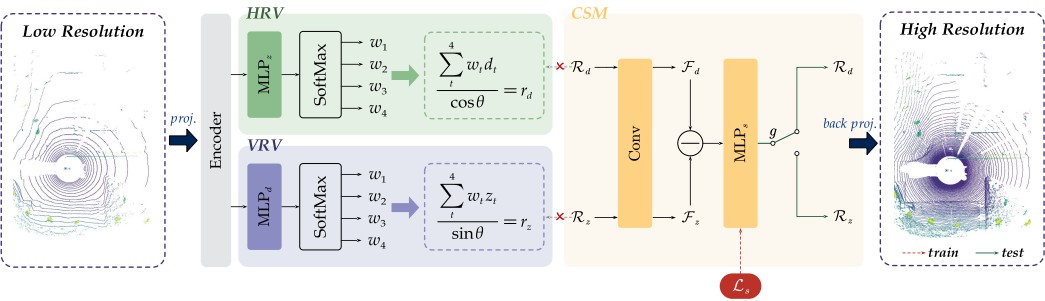

Figure 2: The interpolation framework of WIN. The same local features $\boldsymbol{f}'_{1:4}$ are fed into two independent weight prediction branches to obtain weight values $w_{1:4}$ for the HRV and VRV, respectively. Based on the two upsampled results, we use a shared encoder to extract high-dimensional feature differences to predict the view confidence $g$. The dotted lines and red crosses in the figure indicate that the gradient is not returned here.

upsampling tasks. LiDAR-SR Tian et al. (2022) employs U-Net architecture and transposed convolution to achieve feature extraction and upsampling of range images, and applies MC-Dropout to overcome the edge fuzziness; HALS Eskandar et al. (2022) uses multiple upsampling branches with different receptive fields to deal with the problem of uneven density, and uses virtual normal loss to improve edge distortion; SGSR-Net Chen et al. (2024) combines a CNN model with vertical space and channel attention enhancement , and proposed a structure-guided Monte Carlo filtering to achieve remarkable results on indoor data; Recently, TULIP Yang et al. (2024) is inspired by Swin-IR Liang et al. (2021) and modifies the patch and window geometry of the network to better adapt to LiDAR data. Although explicit methods report considerable accuracy, there is always a need to adapt the network architecture for different input and output resolutions, which makes them inflexible. In addition explicit methods do not directly benefit from sparse geometric information and therefore are usually not geometrically reliable.

**2) Implicit function methods.** Implicit function methods transform the LIDAR upsampling task into learning a continuous function on a 2D image domain. As a result, Implicit function methods are more flexible than explicit methods, and can achieve upsampling of any rate with only one training. LIIF Chen et al. (2021c) first proposed an implicit function for image super-resolution, by predicting the color of a given query point. Inspired by LIIF, ILN Kwon et al. (2022) simulates the scanning of LiDAR by learning weights in the interpolation instead of the values. Thank for the character of convex combination, ILN has achieved great improvement in 3D accuracy. Subsequently, IPN Park et al. (2023) proposed so-called on-the-ray positional embedding to obtain more 3D information. There are some other methods related to LiDAR interpolation, but they focus to increase the frame rate of scanning Zeng et al. (2022); Lu et al. (2021); Liu et al. (2021; 2020). Although these implicit function methods are fast and effective, but they all ignore the limitations of geometric expression of range view when interpolating. In contrast, our method decomposes the range image into two virtual views to better express complex geometry while retaining lightweight and efficiency.

## 3 METHODOLOGY

### 3.1 OVERVIEW

Given a low-resolution LiDAR points $\mathcal{P}_l = \{\boldsymbol{p}_1, \boldsymbol{p}_2, \ldots, \boldsymbol{p}_n\}$ with $n$ points (in which $H, W$ are the vertical and horizontal resolution), our goal is to generate high-resolution ones with $k_1 H \times k_2 W$ resoltion. $k_1, k_2$ are the upsampling factors.

The whole pipeline of WIN is illustrated in Fig. 2. Initially, the point clouds are converted to range images $\mathcal{R}_l$, and then pixel features $\mathcal{F}$ are extracted from $\mathcal{R}_l$. We then feed $\mathcal{F}$ into our variable view interpolation module to predict the neighborhood weights of the query points in each view. We interpolate from the different views based on the predicted weights to obtain high resolution range images $\mathcal{R}_d$ and $\mathcal{R}_z$. To fuse the results of the two views, we input $\mathcal{R}_d$ and $\mathcal{R}_z$ into the CSM and select the best view for each upsampled point by the view confidence scores $\mathcal{G}$ predicted by the

CSM. Finally, we back project the fused range images $\mathcal{R}_h$ to obtain the high-resolution point clouds $\mathcal{P}_h$.

### 3.2 FEATURE EXTRACTION

Firstly, we project LiDAR point clouds onto the range view. The range view projection process can be fomulated as

$$
\begin{bmatrix} v \\ h \end{bmatrix} = \begin{bmatrix} \frac{H}{v_{\max} - v_{\min}} \cdot (\arctan(z/d) - v_{\min}) \\ W \cdot \arctan\left(y/x\right)/(2\pi) \end{bmatrix}, \tag{1}
$$

where $v_{\max}, v_{\min}$ represent the maximum and minimum vertical angles, $(v, h)^\top$ is the image coordinate of point $(x, y, z)^\top$. Then we feed the low-resolution range image $\mathcal{R}_l$ into the backbone network to obtain the set of pixel embeddings $\mathcal{F}$. Consistent with the existing implicit function methods, we use the backbone network EDSR Lim et al. (2017) for feature extraction, which is known for its efficient super-resolution ability. EDSR removes the redundant batch normalization (BN) layers from SRResNet Ledig et al. (2017), which helps preserve the original scale information in the super-resolution task. It is worth noting that our method can be seamlessly integrated into any backbone network.

### 3.3 VARIABLE-VIEW INTERPOLATION MODULE

Existing implicit function methods focus on designing complex interpolation schemes while ignoring the limitations imposed by RV. As shown in Fig. 1, RV is insufficient for describing non-smooth geometric surfaces, leading to the loss of sharp edges during the interpolation process. Unlike these methods, we propose the variable-view interpolation module, which decouples RV into HRV and VRV, the idea behind this is to uncover the distribution pattern of the point cloud through different views and eliminate the distortion caused by spherical projection.

Based on the features $\mathcal{F}$, we need to predict the interpolation weights by utilzie the relative relationship between the query point and its neighboring points. First, for any query point $q$, we find its four nearest neighboring point features $f_{1:4}$ in the image plane. To incorporate the relative position information, we use the relative position of the pixel center to the query ray $\Delta q_t$ to generate a local feature embedding

$$
f' = \mathrm{MLP}_{\mathrm{pos}}(\Delta q) + f, \tag{2}
$$

here the MLP refers to multi-layer perceptrons. With arbitrary encoders, we can predict weights from local features $f'_{1:4}$. We use multi-layer perceptrons(MLP) to improve training and inference speed. Specifically, For each view, the MLP projects the neighborhood embedding

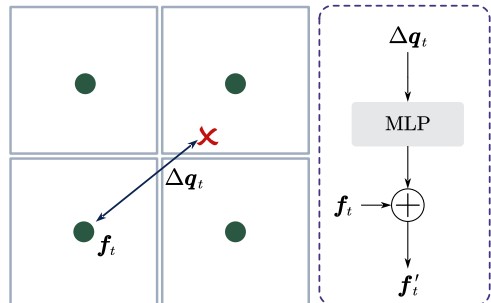

Figure 3: Local embedding module. The figure represents a process embedding the relative position $\Delta q_t$ and extracted features $f$ to compose a local feature embedding $f'_t$.

features $f'_{1:4}$ into a lower-dimensional space, and then the interpolation weights $w_{1:4}$ are computed using the softmax function. Note that our two branches use different MLPs but share the same neighborhood embeddings $f'_{1:4}$. This design allows the variable-view interpolation module to be easily placed behind any encoder and decrease the parameters.

Based on the predicted weight, we project the points to the corresponding views for interpolation respectively. The specific interpolation process of HRV and VRV is expressed by Eq. 3:

$$
\begin{bmatrix} r_d \\ r_z \end{bmatrix} = \begin{bmatrix} \left(\sum_t^4 w_d(f'_t|\theta_d) \cdot d_t\right) / \cos v \\ \left(\sum_t^4 w_z(f'_t|\theta_z) \cdot z_t\right) / \sin v \end{bmatrix}, \tag{3}
$$

where $q = (u, v)$ is the coordinates of a query point, $w(f'_t|\theta)$ is the weight predictor. By using other plane for projection, this algorithm can be extended to arbitrary view. Here, we choose HRV and VRV for their regular distribution and easy transformation.

### 3.4 CONTRAST SELECTION MODULE

To benefit from the results of both views, we designed the contrast selection module(CSM). It predict the better one between HRV and VRV for each query point. We argue that the feature differences are proportional to the view differences, so our CSM predicts the best view by capturing the difference between two results. In addition, we note that the best view is quiet uncertain, so we model the prediction confidence using a Gaussian distribution and set it as the optimization objective for the CSM, encouraging the network to focus on key complementary regions.

Specially, we first use a shared convolutional network to obtain two high-resolution feature maps, $\mathcal{F}_d$ and $\mathcal{F}_z$. Then, we compute the difference between these two feature maps, and use a MLP and Sigmoid function to transform the feature difference into a confidence value $\mathcal{G}$, which range from $[0, 1]$.

$$\mathcal{G} = \text{Sigmoid}\left(\text{MLP}_g\left(|\mathcal{F}_d - \mathcal{F}_z|\right)\right). \tag{4}$$

Then, the final prediciton of WIN can be expressed as

$$\mathcal{R} = \begin{cases} \mathcal{R}_d & \text{where } \mathcal{G} < 1/2, \\ \mathcal{R}_z & \text{where } \mathcal{G} \geq 1/2. \end{cases} \tag{5}$$

However, we need to set a suitable truth value $\hat{\mathcal{G}}$ to supervise the view confidence $\mathcal{G}$ predicted by the CSM. According to our requirements, this truth value should satisfy (i) in the range of $[0, 1]$; (ii) VRV interpolation is chosen for confidence $> 1/2$, otherwise HRV; (iii) as the difference between the two predicted values gets larger, the confidence is closer to $0/1$.

To achieve the desired properties, we use a probabilistic approach to model difference between two views. For a query ray $q$, let the interpolation results from top and side view be $r_d$ and $r_z$, respectively, and the ground truth distance value be $r$. We express the distances of the two predicted values relative to the ground truth with normalized probability values, as follows:

$$\hat{g} = \frac{P(r_z|r)}{P(r_d|r) + P(r_z|r)}. \tag{6}$$

This ensures that $\hat{g}$ remains within the interval $[0, 1]$. If $r_z$ is closer to the true value, then $\hat{g} > \frac{1}{2}$; otherwise, $\hat{g} < \frac{1}{2}$. Assuming $P(\cdot \mid r)$ is a one-dimensional Gaussian distribution with mean $r$ and a standard deviation proportional to the distance, we have:

$$P(r_z|r) = \frac{1}{\sqrt{2\pi}\lambda r}e^{-\frac{(r_z-r)^2}{2(\lambda r)^2}}, P(r_d|r) = \frac{1}{\sqrt{2\pi}\lambda r}e^{-\frac{(r_d-r)^2}{2(\lambda r)^2}}, \tag{7}$$

here, $\lambda r$ represents the standard deviation of the probability distribution, where $\lambda$ is a constant. Substituting this into the formula, we obtain:

$$\begin{aligned} \hat{g} &= e^{-\frac{(r_z-r)^2}{2(\lambda r)^2}} \bigg/ \left(e^{-\frac{(r_z-r)^2}{2(\lambda r)^2}} + e^{-\frac{(r_d-r)^2}{2(\lambda r)^2}}\right) \\ &= \text{SoftMax}\left(-\frac{(r_z-r)^2}{2(\lambda r)^2}, -\frac{(r_d-r)^2}{2(\lambda r)^2}\right)_1, \end{aligned} \tag{8}$$

where subscript 1 represents the first element of the vector.

### 3.5 LOSS FUNCTION

The loss function needed for our network can be divided into two parts: reconstruction loss and selection loss. Where the reconstruction loss we use mean absolute error(MAE) to evaluate the absolute error for each pixel on the range view. For the selection loss part, it is inappropriate to use MAE or MSE, because when $\hat{g}$ is greater than (or less than) $1/2$, we hope that the loss is 0 when the predicted value $g$ is greater than (or less than) $\hat{g}$. Therefore, the selection loss function is expressed in a form as:

$$\mathcal{L}_g = \max\left(0, (\hat{g} - g) \cdot \text{sgn}(2\hat{g} - 1)\right). \tag{9}$$

Here $\text{sgn}$ denotes the sign function. The final loss function is then obtained by directly adding the reconstruction loss and the selection loss:

$$\mathcal{L} = \mathcal{L}_d + \mathcal{L}_z + \mathcal{L}_g. \tag{10}$$

It's worth to notice that we interrupt the propagation of gradients at $\mathcal{R}_d$ and $\mathcal{R}_z$ , aiming to prevent the gradients of CSM from being propagated back to the encoder, thereby ensuring the training stability of the interpolation network. Even so, our network can still be trained end-to-end.

## 4 EXPERIMENTAL RESULTS

### 4.1 EXPERIMENT SETTINGS

**Datasets**: In our experiments, we included both real and synthetic datasets. (i) For synthetic datasets, we use a virtual dataset built with CARLA simulator Dosovitskiy et al. (2017), followed by TULIPYang et al. (2024) and ILN Kwon et al. (2022). The virtual data capture noise-free point clouds with a vertical FoV of $30°$. We use a (20699/2618) train/test split. (ii) For real-world datasets, we use KITTI. The KITTI dataset Geiger et al. (2012) was obtained using a Velodyne HDL-64E Li-DAR with a vertical FoV of $26.8°$ and a resolution of $64 \times 1024$. We sampled frames randomly from sequences of *2011_10_03* for test, and train all models with other sequences.

It is worth to notice that adjacent frames of LiDAR data usually have similar scene structures, we select point cloud sequences that have no spatial overlap with the training data as test data to more accurately compare the upsampling effect of the models.

**Implementation Details**: In all experiments, the initial learning rate is set as 1e-4 and decayed with a rate of 0.5 for every 50 epoch. Our optimizer is chosen as the Adam optimizer Kingma (2014). The models are implemented using the PyTorch framework Paszke et al. (2019) and run on an Nvidia RTX 4090 GPU with 24GB of memory.

**Comparison Methods**: To demonstrate the effectiveness of our network, We compare it with several LIDAR upsampling networks, where explicit methods include LiDAR-SR Shan et al. (2020) and TULIP Yang et al. (2024), implicit function methods include LIIF Chen et al. (2021c) and ILN Kwon et al. (2022).

**Evaluation metrics**: We use two commonly used metrics: MAE and Intersection-over-Union (IoU). We compute MAE of all pixels in the generated two-dimensional range images. For IoU, we voxelize the point cloud with a voxel size of 0.1m. A voxel is classified as an occupied voxel for each point cloud if it contains at least one point. We then compute the IoU based on the occupancy rate.

### 4.2 SINGLE UPSAMPLING SCALE

We use the same experimental setup and dataset as TULIP and report the precision reproduced by TULIP. It is worth noting that TULIP's reproduction of ILN is not accurate. After discussions with the original author, we retrained ILN for the experiments with single upsampling scale. Specific modifications can be seen in the Supplementary material. Furthermore, we consider that the projection center of KITTI data is not unique, leading to a large number of null shortcomings under the range view. This will lead to unreasonable losses and does not reflect the true performance of the models. For this reason, we adjusted the KITTI projection with reference to Fan et al. (2021). We retrained and evaluated all methods under the same settings.

As can be seen from the Tab. 1 , our WIN achieves the best performance on the CARLA dataset with only 1.7M parameters, where the MAE improves by 4.53% and the IoU improves by 7.01%. This is due to the fact that we take into account the close connection between the local geometry and the interpolated view, and benefit from it through the design of the CSM.

Since the projection center of KITTI is not unique, it makes the neighborhood geometry relationship under the distance view broken. This systematic error from KITTI places higher demands on the model's ability to maintain geometric structure. We take advantage of the complementarity of variable views, making our method far superior to existing methods in IOU. It is worth noticing that, LIIF regresses the distance values directly from the features, making it easy to fit in this incorrect setting, thus resulting in an optimal MAE.

In addition, we also conducted qualitative experimental comparisons. Figure 4 shows the visualization effects of TULIP, ILN and WIN on CARLA dataset. Compared with explicit method (TULIP) method, our WIN can robustly reconstruct 3D surfaces, with less noise and artifacts, and maintain

Table 1: Quantitative experimental results with a single upsampling rate. We choose the virtual dataset created by CARLA simulator and the real-world dataset KITTI raw to evaluate related methods.

| Method | Param(M) | CARLA(32→128) | | KITTI(16→64) | |
|---|---|---|---|---|---|
| | | MAE↓ | IoU↑ | MAE↓ | IoU↑ |
| LiDAR-SR | 34.6 | 0.8216 | 0.2581 | 0.6956 | 0.1470 |
| TULIP | 27.1 | 0.7699 | 0.5152 | 0.6794 | 0.4044 |
| Bilinear | 0 | 1.8128 | 0.1382 | 1.6668 | 0.1541 |
| LIIF | 1.4 | 0.8064 | 0.3502 | **0.6405** | 0.3958 |
| ILN | 1.3 | 0.7613 | 0.5621 | 0.6869 | 0.4024 |
| WIN(Ours) | 1.7 | **0.7268** | **0.6015** | 0.6799 | **0.4189** |

Table 2: Quantitative experimental results with different upsampling rates. We choose the virtual dataset created by CARLA simulator. The resolution of input data is $16 \times 1024$.

| Method | 64×1024 | | 128×2048 | | 256×4096 | |
|---|---|---|---|---|---|---|
| | MAE↓ | IoU↑ | MAE↓ | IoU↑ | MAE↓ | IoU↑ |
| LiDAR-SR | 1.5600 | 0.2782 | 1.7460 | 0.1610 | 1.7530 | 0.1270 |
| TULIP | 1.4776 | 0.3471 | 1.5422 | 0.3451 | 1.5984 | 0.2523 |
| Bilinear | 2.3720 | 0.2020 | 2.5910 | 0.1650 | 2.6460 | 0.1630 |
| LIIF | 1.5388 | 0.2713 | 1.6890 | 0.2480 | 1.7370 | 0.2130 |
| ILN | 1.4168 | 0.3927 | 1.5368 | 0.3476 | 1.6088 | 0.2653 |
| WIN(Ours) | **1.3834** | **0.4561** | **1.4971** | **0.3995** | **1.5698** | **0.2732** |

geometric characteristics. This is because we interpolate from a suitable view, enhancing the geometric reliability of the upsampling process; compared with implicit function method (ILN), our WIN overcomes the limitations of interpolation, thus recovering better in sparse areas. At the same time, WIN retains more fine-grained geometric features. The qualitative results fully indicate that the previous analysis is reasonable, that is, the complementarity of the two views can help upsampling more refinedly, and eliminates geometric distortions introduced by range view projection.

### 4.3 MULTIPLE UPSAMPLING SCALES

Compared with the explicit methods, the core advantage of the implicit function methods is that the upsampling scale can be flexibly adjusted. Therefore, in order to evaluate the adaptability of the models at different upsampling scales, we conducted experiments with multiple upsampling scales on the CARLA dataset. The CARLA dataset containing four different resolutions: $16 \times 1024$, $64 \times 1024$, $128 \times 2048$, and $256 \times 4096$. We use the $16 \times 1024$ resolution as the input and the others as the target resolutions.

Tab. 2 reports the quantitative performances. Our method maintains the geometric reliability of interpolation while flexibly obtaining information from different views, thus achieving the best MAE and IoU. Moreover, as the target resolution increases, our method has an increasing advantage over existing methods. This is due to the fact that the increasing target resolution makes the distance view more limiting. Our WIN, on the other hand, dissolves this limitation by decoupling the range view into HRV and VRV, and thus exhibits more significant advantages at very high upsampling scales.

### 4.4 DOWNSTREAM TASK

In order to realistically evaluate the performance of the upsampling methods, we choose another 3D geometry base task (depth completion) to compare the model's contribution on the downstream task. Specifically, we downsample the depth map in the KITTI depth dataset to 16 lines to simulate the sparse point cloud. We then perform $4 \times$ upsampling using all models pre-trained on KITTI raw Geiger et al. (2012) to reconstruct a 64-line point cloud. We use the reconstructed point cloud for depth completion and compare it with ground truth. For validation metrics, we choose average

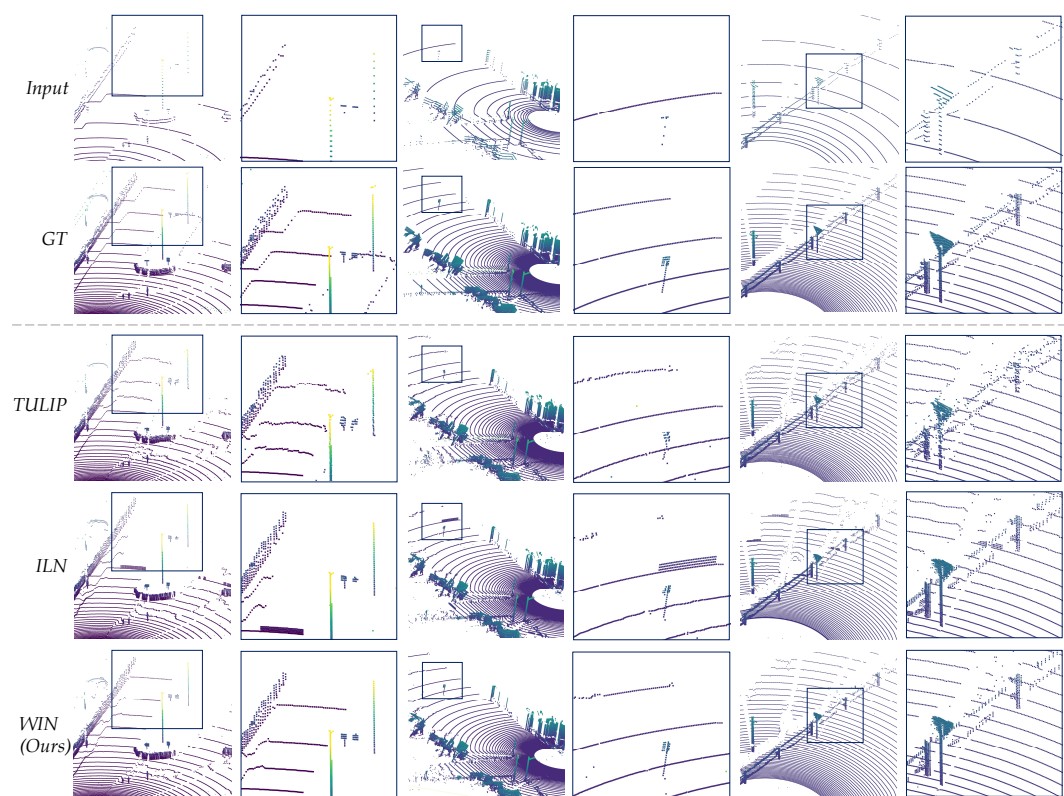

Figure 4: Qualitative results of LIDAR super-resolution obtained by different methods. The area in the red box on the left is shown enlarged in the right panel.

Table 3: Downstream mission performance comparison. We use the KITTI dataset and choose depth completion for comparison. Depth Completion is a geometry foundation task, we use RMSE and MAE for evaluation.

| Method | MAE(mm)↓ | RMSE(mm)↓ |
|---|---|---|
| Low-resolution | 1131.84 | 3470.70 |
| LiDAR-SR | 765.14 | 2630.86 |
| TULIP | 780.76 | 2796.88 |
| LIIF | 671.39 | 2509.77 |
| ILN | 686.52 | 2529.30 |
| WIN(Ours) | **659.18** | **2408.20** |

absolute baseline (MAE) and root mean square baseline (RMSE), which are calculated over all valid pixels.

The result of downstream task are shown in Tab. 3. It can be seen that basically all upsampling methods achieve the enhancement of downstream tasks. Our WIN reconstructs high-resolution point clouds with more accurate geometric properties by expressing richer geometries from different views, thus achieving a significant advantage in both MAE and RMSE. This suggests that WIN may have the greatest potential for application of any method.

## 4.5 ABLATION STUDY

To illustrate the effectiveness of every design proposed in WIN, we conducted ablation experiments on CARLA and KITTI by removing each component as follows. (i) Removing Variable-View in favor of range image for interpolation, the overall model will degenerate into ILN. (ii) Removing the

Table 4: Ablation study. The evaluation results of different components of WIN. We conducted experiments on CRALA and KITTI.

| Method | CARLA(32→128) | | KITTI(16→64) | |
|---|---|---|---|---|
| | MAE↓ | IoU↑ | MAE↓ | IoU↑ |
| w/o Variable-view | 0.7613 | 0.5621 | 0.6869 | 0.4024 |
| w/o CSM($d$) | 0.7559 | 0.5836 | 0.6849 | 0.4082 |
| w/o CSM($z$) | 0.7711 | 0.5601 | 0.6992 | 0.4033 |
| w/o $\hat{g}(d+z)$ | 0.7313 | 0.5905 | 0.7123 | 0.3703 |
| WIN($d+z$) | **0.7268** | **0.6015** | **0.6799** | **0.4189** |

Contrast Selection Module, we report the interpolation results of the two different views separately. (iii) Removing the confidence score constraint, we use binary classification loss as a replacement.

It can be seen from table 4 that our full pipeline obtains the lowest MAE and IOU values, removing any component from it will lead to a degradation in the network. On MAE and IoU, the CSM achieved an average improvement of 2.3864% and 2.9459% respectively. It's interesting that in all experiments, using distance for interpolation is less effective than using planar distance. It is due to the fact that the range view contains more shape distortion,while the other two views alleviate this problem through orthogonal projection.

And, the network supervised by our confidence values $\hat{g}$ achieve a better performance than binary cross-entropy. It's due to the fact that we specify the output of the selection task as the confidence level of the view in contrast to classification label. This allows us to ignore the loss due to the constant change of classification labels

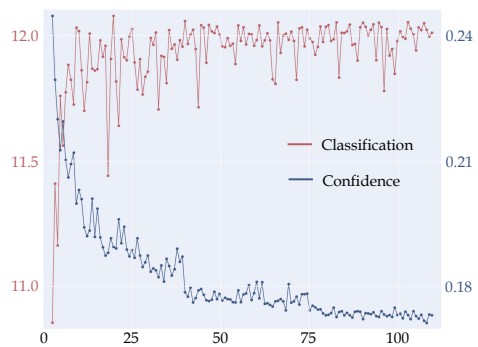

Figure 5: The selection loss curve of the network during training. The dataset is KITTI.

during training and instead focus on regions where the gap between the two views is large. As shown in Fig. 5, We compare the loss curves of the network under different supervision methods. The selection loss supervised by binary cross entropy rises rapidly and tends to be invariant, while our CSM gradually converges by the confidence score constraint method. This shows that the strategy of constraining the view selection process by predicting the confidence score allows us to achieve more effective upsampling.

## 5 CONCLUSION

In this paper, we design a LiDAR upsampling network called WIN, which achieves the effect of varying the interpolation view with geometric properties by cleverly incorporating different view representations into the distance view. For the first time, we decouple the representation of range view into two orthogonal view representations, HRV and VRV, and implement learnable implicit interpolation for each. We also present a contrast selection module that benefits from different views by probabilistically modeling the confidence level of the predicted views. We evaluate our approach on both large-scale synthetic datasets and real-world datasets. The experimental results show that our WIN significantly improves the interpolation accuracy, especially the geometric accuracy. At the same time WIN occupies almost minimal memory. In addition, our method exhibits the best performance in a downstream task (depth completion).

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

# A  APPENDIX

## A.1  COMPLEMENTARY OF HRV AND VRV

In order to more intuitively illustrate the distribution of HRV and VRV in different areas, we selected some samples for visualization, as shown in Fig. 6. Combined with the distance map, it is easy to find that the distribution is actually closely related to the geometry of the object surface. For example, VRV is more advantageous on flat ground, while HRV performs better on the surface of the object.

## A.2  VISUALIZATION OF DEPTH COMPLETION

Here we present a qualitative evaluation of depth completion experiment. Observing the distribution of the error map, we can see that our WIN is more accurate in scene structure and produces smaller errors. We also observe that TULIP produces significant errors in flat regions, which may be due to the lack of geometric reliability of feature regression-based methods. Compared with ILN, because we alleviate the limitations of the interpolation method, smaller errors are produced in most areas.

## A.3  OBJECT DETECTION

Object detection is also a important downstream task of point cloud. Similar to the experimental setup of depth completion, we upsample the 16-line point clouds to match the raw data, directly use a model pre-trained on KITTI Object Dataset and generate the 3D bounding boxes on the generated, ground truth and low-resolution point clouds. We apply PointPillar Lang et al. (2019), which is commonly used for object detection. Since existing methods do not support upsampling of intensity values, we perform nearest neighbor interpolation in 3D space to compensate for the lost intensity information. The final results are shown in Tab. 5, The accuracy of other methods are reported by

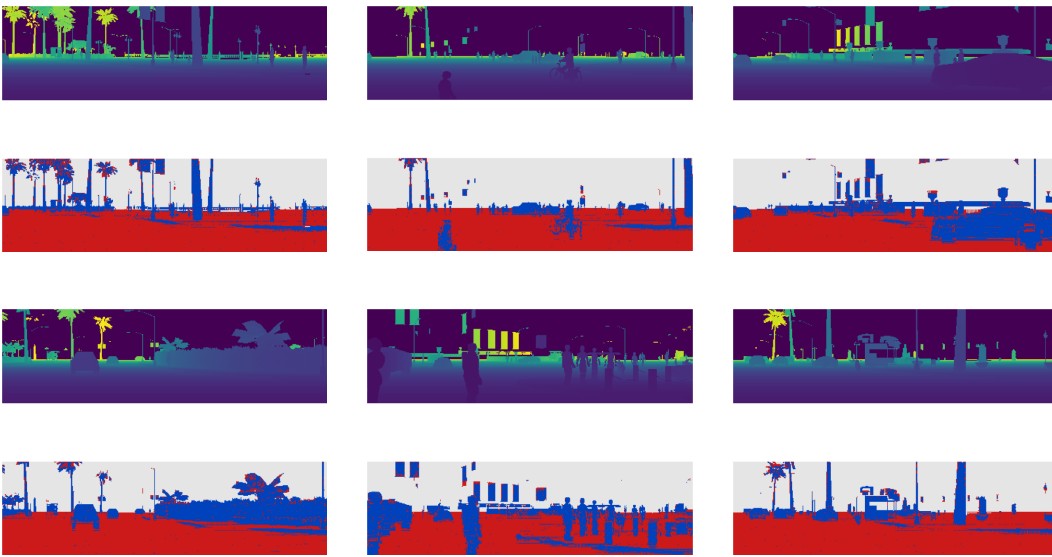

Figure 6: Visulization of the complementary of HRV and VRV, where blue means HRV is better and red means VRV is better.

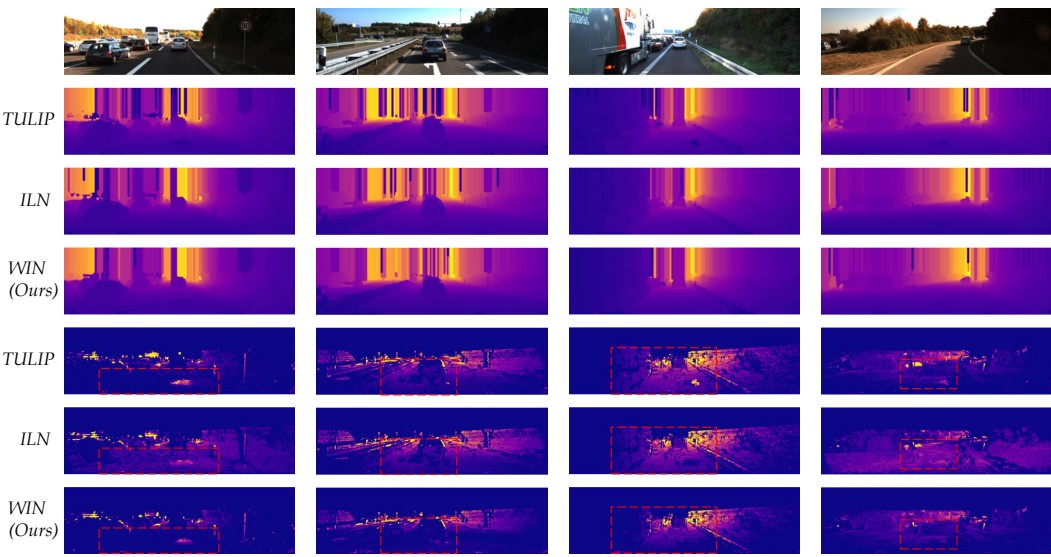

Figure 7: Qualitative results of depth completion using upsampled point clouds. The bottom three rows of error map fully illustrate that our WIN recovers the scene geometry more accurately.

Yang et al. (2024). The results show that our method outperforms ILN and achieves a significant improvement based on low-resolution point cloud. We also compared with the metrics reported by TULIP, although we were unable to reproduce similar results.

## A.4 DISCUSSION

**Is it better to use a weighted sum to fuse the HRV and VRV?** Although using weighted sum to fuse the two views is a natural idea since it makes the model easier to train and the theoretical interpolation range larger (covering the interpolation range of HRV and VRV), we believe that it destroys the geometric reliability of the interpolation algorithm. To rigorously verify this idea, we retrained the model using a weighted sum. Note that at this point we no longer need to choose the

Table 5: Object detection performance comparison. We evaluate pretrained PointPillar on point clouds upsampled by different methods and report the overall results (averaged over classes 'Car', 'Cyclist' and 'Pedestrian').

| Method | Easy | Moderate | Hard |
|---|---|---|---|
| Low-resolution | 10.05 | 9.03 | 8.8 |
| LiDAR-SR | 29.27 | 24.15 | 20.39 |
| TULIP | 50.23 | 37.57 | 32.12 |
| ILN | 38.29 | 28.61 | 23.67 |
| WIN(Ours) | 40.11 | 28.80 | 26.67 |

Table 6: Quantitative experimental results with different upsampling rates. We choose the virtual dataset created by CARLA simulator. The resolution of input data is $16 \times 1024$. We abbreviate the weighted sum method as WS.

| Method | 64×1024 | | 128×2048 | | 256×4096 | |
|---|---|---|---|---|---|---|
| | MAE↓ | IoU↑ | MAE↓ | IoU↑ | MAE↓ | IoU↑ |
| ILN | 1.4168 | 0.3927 | 1.5368 | 0.3476 | 1.6088 | 0.2653 |
| WS | 1.4131 | 0.3604 | 1.5004 | 0.4061 | 1.7218 | 0.1938 |
| WIN(Ours) | **1.3834** | **0.4561** | **1.4971** | **0.3995** | **1.5698** | **0.2732** |

loss $\mathcal{L}_g$ since the weights have been explicitly guided. The comparison on CARLA dataset is shown in Tab. 6. The results show that the weighted sum cannot even achieve better results than ILN, probably because the results are too dependent on the weights. On the other hand, the weighted sum model shows a significant decrease in IoU, which we believe supports the conjecture that the weighted sum geometry is insufficiently reliable.

