# OpenReview forum: "WIN: Variable-View Implicit LIDAR Upsampling Network"
_ICLR.cc/2025/Conference — Submitted to ICLR 2025_

### Official Review · Reviewer_Gpfk · 2024-10-31

**Soundness:** 3
**Presentation:** 3
**Contribution:** 3
**Rating:** 3
**Confidence:** 4

**Summary:**

They propose a novel Variable-View Implicit LiDAR upsampling Network (WIN), which decouples the 3D representation of Range View (RV) into two novel virtual view representations Horizon Range View (HRV) and Vertical Range View (VRV).

1. A reason of Variable-View upsampling is discussed. HRV and VRV, as an orthogonal transformation of RV, can provide more perspectives for observation without losing any geometric information.

2. Implicit function methods are used to interpolate points in different views. It allows us to enjoy the advantages of variable-view representations without introducing unnecessary parameters or changing network architecture.

3. A contrast selection module is designed to help each interpolation point to choose the corresponding best view based on the geometric differences in the upsampled image from different representations.

4. Model the best view selection process as a probability distribution problem.

5. Extensive experiments shows that, WIN achieved SOTA performance on both virtual and real-world datasets. An improvement of 4.53% and 7.01% is achieved on MAE and IoU, respectively.

**Strengths:**

Strengths:

1. Method is easy to follow and understand.
2. Figure 1 provides a clear insight, which makes readers better understand the motivation.

**Weaknesses:**

Weaknesses:

1. The writting in Introduction is overpacking. In introduction section, it seems that the proposed method is complex with theoretically meaning. But, if we read the content in Method section, we find the proposed method is not complex. For example, the authors claim that they model the best view selection process as a probability distribution problem. In Sec. 3.4, Eqs. (6)-(8) is not impressive.

2. Few state-of-the-art methods are compared in this manuscipt. And it is recommanded to add more comparision results.

3. Is it really essential to decompose LiDAR point cloud with HRV and VRV? Range view is enough. In range view, the horizontal and vertical information can be used together, achieving more accurate results.

4. Lacks of theoretical novelty. ICLR is a conference that focuses more on learning theory and related topics.

**Questions:**

Questions:

Is it really essential to decompose LiDAR point cloud with HRV and VRV? Range view is enough. In range view, the horizontal and vertical information can be used together, achieving more accurate results.

Lacks of theoretical novelty. ICLR is a conference that focuses more on learning theory and related topics.

---

> ### Author Response · Authors · 2024-11-28
>
> We sincerely thank Reviewer `Gpfk` for the valuable comments provided during this review. The point-to-point responses are as follows.
>
> ---
> > **W1**: The writting in Introduction is overpacking...the proposed method is not complex.
>
> **A**: Thank you for your suggestion. Although our motivations and methods are simple, each of our modules is thoroughly considered and designed. First, we study past methods and point out the advantages and limitations of interpolation-based methods. We then visually show how to interpolate from different views (which, to our knowledge, has not been explored before) and point out the limitations of a single view. Considering the scene prior, we choose horizontal range view and vertical range view, and hope to fuse the two. In order to reasonably merge the results under the two views, we propose a selection module. Finally, in order to stably train the selection module, we formulate the loss function as needed (L287-290).
>
> We further justify these designs. Regarding the fusion approach, we draw on other reviewers' comments and compare it with the weighted sum approach. The results show that the weighted sum does not perform as well as our chosen module. Regarding the loss function, we have demonstrated in ablation experiments that it not only correctly guides view selection (502-513), but also improves the stability of the algorithm (L514-525).
>
> ---
>
> > **W2**: Few state-of-the-art methods are compared in this manuscipt. And it is recommanded to add more comparision results.
>
> **A**: Thanks for the comment, but we clarify that we compared all state-of-the-art scene-level point cloud upsampling methods. To dispel your doubts, we compare with recent object-level method, and the results are shown in Table 1. The results show that **our method far outperforms [1] because [1] cannot be adapted to LiDAR data**.
>
> |  Methods  |   MAE(m)   |    IoU     | Time(sec) |
> | :-------: | :--------: | :--------: | :-------: |
> |   PUDM    |  12.2957   |   0.0713   |   4.75    |
> | WIN(Ours) | **1.3834** | **0.4561** | **0.03**  |
>
> We also added the visualization results of depth completion in the Appendix A.2, which show that our method yields point clouds that are closer to the real geometry.
>
> ---
>
> > **W3**: Is it really essential to decompose LiDAR point cloud with HRV and VRV? Range view is enough.
>
> **A**: The core point of our article is that the range view, although it contains all the information, can be limited in the interpolation framework**. We visualize the limitations of the range view in Figure 1. This limitation does not stem from a lack of information, but rather from the fact that the interpolated points are restricted to convex combinations of neighborhoods. This restriction provides a good prior in most scenarios, making the network easy to optimize. **However, on non-smooth surfaces, interpolation from a single view may lead to theoretical infeasibility**. For this reason, we propose to upsample from the horizontal distance view (HRV) and vertical distance view (VRV) separately, which not only helps to alleviate the limitation, but also naturally incorporates an understanding of the scene's geometric distribution: i.e., near-horizontal for the ground and near-vertical for objects and façades. All these factors make our approach go beyond existing methods and allow for a more accurate understanding of the geometric features in the scene.
>
> If you are interested in the distribution of HRV and VRV, we visualize it in A.1, which intuitively shows the complementary of the two views, and gives direct evidence that using multiple views improves performance.
>
> ---
>
> > **W4**: Lacks of theoretical novelty.
>
> **A**: We expected to explain our work in a more theoretical way too, but unfortunately the real world geometry is quite complex. At the same time, it seems too simple to prove that the interpolation range of multiple views is greater than that of a single view. Therefore, in the paper, we prefer to illustrate the motivation of our approach in a more intuitive way as well as through experimental results, cf. Fig. 1.
>
> ---
>
> To summarize, we have designed an effective LiDAR upsampling framework based on an intuitive motivation. Our components are simple but carefully designed, and we justify our designs by extensive experiments. The theoretical supplement will be our next work. If you have any other doubts, feel free to point them out to us further.
>
> Last but not least, we thank you again for your comments. We hope our answers solve your doubts.
>
>
> [1] Qu W, Shao Y, Meng L, et al. A Conditional Denoising Diffusion Probabilistic Model for Point Cloud Upsampling[C]//Proceedings of the IEEE/CVF Conference on Computer Vision and Pattern Recognition. 2024: 20786-20795.

---

> > ### Comment · Reviewer_Gpfk · 2024-11-30
> > **Feedback to authors**
> >
> > Thank you very well for your response. I strongly encourage the authors to submit their work to Robotics related journal or conference, such as ICRA/IROS/RAL. If they submit their work to these conferences, there must be a strong accept. ICLR, however, focuses on the theoretical contribution on representation learning.

---

### Official Review · Reviewer_Lkn5 · 2024-11-07

**Soundness:** 3
**Presentation:** 2
**Contribution:** 2
**Rating:** 6
**Confidence:** 4

**Summary:**

This work proposes a Variable-View Implicit lidar upsampling Network (WIN) to overcome the limitation of previous work only using a single perspective. In particular, this work first decouples the range view into two novel virtual view representations: Horizon Range View and Vertical Range View, to compensate for the missing geometric information. Then, a Contrast Selection Module is designed to guide the selection process by capturing the feature differences between the different representations. Experimental results demonstrate the effectiveness of the proposed WIN across different datasets and downstream vision tasks, such as depth completion.

**Strengths:**

+ The motivation for decoupling the 3D representation of a ranging image into two orthogonal view representations is clear and sound. It well addresses the limitation of previous works which only leverage a single perspective view.
+ The interpolation performance is promising and the proposed method can also facilitate some downstream tasks.

**Weaknesses:**

- While two orthogonal view representations can compensate for more information than the single perspective view in this paper, such a strategy (jointly considering the Horizon Range View and Vertical Range View) has been widely explored in previous 3D vision works. For example, "Joint 3D Proposal Generation and Object Detection from View Aggregation", "Multi-View 3D Object Detection Network for Autonomous Driving", "Deep Continuous Fusion for Multi-Sensor 3D Object Detection", etc. The authors should highlight their unique contributions and insights compared to these cross-view 3D works.
- In the Introduction, the contents of line-120 to line-127 and line-130 to line-138 are redundant with each other. The authors are suggested to integrate these two parts.
-  The proposed contrast selection module can help each interpolation point to choose the corresponding best view. Would this module be further improved using the feature fusion strategy across different views? The concrete reason for selection rather than fusion applied in this work needs to be clarified.
- Some key works about the LiDAR point cloud interpolation-related tasks are missing. For example, "Plin: A network for pseudo-lidar point cloud interpolation", "Pseudo-lidar point cloud interpolation based on 3d motion representation and spatial supervision", "PointINet: Point Cloud Frame Interpolation Network", "IDEA-Net: Dynamic 3D Point Cloud Interpolation via Deep Embedding Alignment", etc. The reviewer suggests that the authors include these references and discuss them briefly in the related work or introduction sections.
- The experimental results are insufficient. For example, this paper lacks important qualitative evaluations of the ablation study and depth completion comparison.
- This paper is less polished and shows some poor presentations. For example, in line 210: r1, r2 is the upsampling factors; in line 279: use a MLP; in line 272: module(CSM), etc. Besides, the predicted view confidence g is missing in Figure 2.

**Questions:**

Can the proposed WIN help more downstream LiDAR-based works? Could you provide more discussions?

---

> ### Author Response · Authors · 2024-11-25
>
> We sincerely thank Reviewer `Lkn5` for the valuable comments provided during this review. The point-to-point responses are as follows.
>
> ---
>
> > **W1**: The authors should highlight their unique contributions and insights compared to these cross-view 3D works.
>
> **A**: As you said, multi-view approaches have been explored a lot and they are impressive. But specifically, applying multiple views in the LiDAR upsampling framework is not easy, and requires more complex network design and fusion modules to achieve fine-grained feature extraction. We analyze the limitations of interpolation, go a step further to show the necessity of multiple views in the up-sampling task, and implement a new multi-view interpolation framework with a small number of parameters.
>
> Previous methods usually extract features from multiple views and fuse them, but do not explicitly exploit the data distribution under multiple views. Different with previous work, **Our method is essentially a new upsampler** , which is independent with feature extraction.
>
> ---
>
> > **W2** : In the Introduction, the contents of line-120 to line-127 and line-130 to line-138 are redundant with each other.
>
> **A**: Thank you for your suggestions, we have revised the relevant parts in the updated paper.
>
> ---
>
> > **W3**: Would this module be further improved using the feature fusion strategy across different views? The concrete reason for selection rather than fusion applied in this work needs to be clarified.
>
> **A**: Regarding the fusion method, our motivation is simple: we do not want to use the method of feature fusion and then predict range, because we have studied past work and found that even if we introduce several times more parameters and more complex structures like [1], it cannot superior simple interpolation methods. This is because the interpolation methods makes full use of known geometric information and has stronger generalization capabilities. In summary, **we hope to retain the advantages of the interpolation algorithm**.
>
> After that, the key question is how to fuse the two predictions. Another reviewer `Xvya` had a related query, arguing that weighted averaging was more intuitive. However, in practice weighted averaging prevents both branches from being fully optimized and the final accuracy is not ideal. We show the comparison result on CARLA dataset in the table below.
>
> **Table1**: Comparison of weighted sum and selection.
>
> |   Methods    | MAE(m) |  IoU   |
> | :----------: | :----: | :----: |
> | weighted sum | 1.4131 | 0.3604 |
> |     Ours     | 1.3834 | 0.4561 |
>
> ---
>
> > **W4** : Some key works about the LiDAR point cloud interpolation-related tasks are missing.
>
> **A**: Thanks for the reminder, although these methods are less relevant to our task, we have added references to these works in the updated paper.
>
> ---
>
> > **W5**: The experimental results are insufficient. For example, this paper lacks important qualitative evaluations of the ablation study and depth completion comparison.
>
> **A**: We add the visualization results of depth completion in the A.2, which show that **our method yields point clouds that are closer to the real geometry**. We also provide **visualizations of the complementarity of the two views** to further justify our motivation in A.1. Our experiments provide a comprehensive comparison of related methods, which is sufficient to demonstrate the effectiveness of the proposed method.
>
> ---
>
> > **W6**: This paper is less polished and shows some poor presentations. For example, in line 210: r1, r2 is the upsampling factors; in line 279: use a MLP; in line 272: module(CSM), etc. Besides, the predicted view confidence g is missing in Figure 2.
>
> **A**: Thank you for pointing out the typos in our manuscript. We have corrected them in the updated paper.
>
> ---
>
> > **Q1**: Can the proposed WIN help more downstream LiDAR-based works?
>
> **A**: We add experiments on object detection and compare the performance of different methods in A.3. The results in the Table 2 show that our **WIN can effectively improve the object detection result by upsample the low-resolution point clouds.**
>
> **Table2**: Results of object detection performance by different upsampling methods.
>
> |  Method   |   Easy    | Moderate  |   Hard    |
> | :-------: | :-------: | :-------: | :-------: |
> |  16-line  |   10.05   |   9.03    |    8.8    |
> |    ILN    |   38.29   |   28.61   |   23.67   |
> | WIN(Ours) | **40.11** | **28.80** | **26.67** |
>
> ---
>
> [1] Yang B, Pfreundschuh P, Siegwart R, et al. TULIP: Transformer for Upsampling of LiDAR Point Clouds[C]//Proceedings of the IEEE/CVF Conference on Computer Vision and Pattern Recognition. 2024: 15354-15364.

---

> > ### Comment · Reviewer_Lkn5 · 2024-11-27
> >
> > Thanks for your response, which addressed some of my concerns. However, it is not easy to find the revision parts regarding previous concerns in the updated paper. Could you highlight where you specifically revised the paper, such as W1, W2, W3, and W4?

---

> > > ### Author Response · Authors · 2024-11-27
> > >
> > > Thank you for your reply, we are truly delighted to have addressed your concerns. We have uploaded a new PDF and highlight the modified part of it into green. Specifically, The modified part of each weakness is:
> > >
> > > - **W1**: we had illustrated this before in **L102-106**.
> > > - **W2**: we have reorganized the language, see **L120-125**.
> > > - **W3**: we discuss weighted fusion versus selection in **L750-782**.
> > > - **W4**: we added citation of these works in **L198-199**.
> > > - **W5**: we provide qualitative analysis of view complementarity and downstream task, see **L680-744** for details.
> > >
> > > We sincerely thank you for your contributions to improving the quality of this work.

---

> > > > ### Comment · Reviewer_Lkn5 · 2024-11-28
> > > >
> > > > Thanks for your quick update. Considering most of my concerns have been addressed, I would like to increase my rating. However, the authors still need to make great efforts to solve the concerns raised by other reviewers. I also wish other reviewers could participate in the rebuttal process. Thanks.

---

### Official Review · Reviewer_Qv6x · 2024-11-07

**Soundness:** 2
**Presentation:** 3
**Contribution:** 2
**Rating:** 5
**Confidence:** 3

**Summary:**

This paper proposes WIN, a variable-view implicit LiDAR upsampling network, providing a holistic understanding of the scene geometry. Different from previous work relying on single range view representation, this paper incorporates two decomposed views, Horizon Range View (HRV) and Vertical Range View (VRV), to achieve more accurate interpolation results. Furthermore, the authors employ a Contrast Selection Module (CSM) with the view confidence score to predict the best point from two orthogonal views. Based on this, WIN finally outperforms existing methods on both upsampling and downstream tasks.

**Strengths:**

- In contrast to previous approaches that rely only on a single range view, the authors of this paper provide a novel idea of decomposing it into two orthogonal views to better solve the LiDAR upsampling problem.

- The interpolation module is based on a local implicit function, which is more flexible to adapt to different views. And the overall model is lightweight and efficient.

**Weaknesses:**

- This paper actually converts the interpolation of depth on range view to the interpolation of 3D coordinates. However, the interpolation on the two orthogonal views is independent of each other and produces inconsistent results.  Although the paper uses another module to select one of the points, it may not be a particularly plausible way of fusion.

- Interpolation using only a simple MLP may be difficult to process in different regions. Especially when the encoder features are only from the range view, the model may not possess the perception ability in HRV and VRV viewpoints.

**Questions:**

Please refer to the Weaknesses, and there are additional concerns as follows:

- As mentioned in L237, "RV is insufficient for describing non-smooth geometric surfaces", but the projection of HRV and VRV does not change its essence and still suffers from this problem.

- When choosing the four nearest neighbors of a query point, how to solve the interpolation problem for blank areas, occlusions, or object edges?

- The argument for predicting the best point through feature differences lacks a theoretical basis or a more in-depth explanation.

- In the selection module, the ground truth is modeled using a probabilistic approach. Is it possible to lead to a random distribution? What do the final distributions and proportions from the HRV and VRV look like? It's better to provide more visualization. Also, as opposed to this discrete selection, why not consider other fusion ways like weighted averaging or learnable fusion?

Other minor issues:

- L208: Actually, the number of points in the LiDAR point cloud is not equal to n = H × W points.

- L216: typo for $R_{l}$.

---

> ### Author Response · Authors · 2024-11-25
>
> We sincerely thank Reviewer `Qv6x` for the valuable comments provided during this review. The point-to-point responses are as follows.
>
> ---
>
> > **W1**: (Motivation of Selection Module) The interpolation on the two orthogonal views is independent of each other and produces inconsistent results. Although the paper uses another module to select one of the points, it may not be a particularly plausible way of fusion.
>
> **A**: The interpolated results of the two views are both geometrically intuitive, so most of the time the two results are both reasonable. And the key to the success of our method lies in the fact that **we capture the widespread inconsistency between the two views**.
>
> Regarding the fusion method, our motivation is simple: we do not want to use the method of feature fusion and then regressing range, because we have studied past work and found that even if we introduce several times more parameters and more complex structures like [1], it cannot superior simple interpolation methods. This is because the interpolation methods makes full use of known geometric information and has stronger generalization capabilities. In summary, **we hope to retain the advantages of the interpolation algorithm.**
>
> After that, the key question is how to fuse the two predictions. Another reviewer `Xvya` had a related query, arguing that weighted averaging was more intuitive. However, in practice weighted averaging prevents both branches from being fully optimized and the final accuracy is not ideal. We show the comparison result on CARLA dataset in the table below.
>
> **Table1**: Comparison of weighted sum and selection.
>
> |     Methods     |   MAE(m)   |    IoU     |
> | :-------------: | :--------: | :--------: |
> |       ILN       |   1.4168   |   0.3927   |
> |  weighted sum   |   1.4131   |   0.3604   |
> | Selection(Ours) | **1.3834** | **0.4561** |
>
> ---
>
> > **W2**: Interpolation using only a simple MLP may be difficult to process in different regions. Especially when the encoder features are only from the range view, the model may not possess the perception ability in HRV and VRV viewpoints.
>
> **A**: Predicting interpolated weights is essentially predicting the similarity of a query point to neighboring points. ILN uses self-attention to learn relative relationships, but results in large matrix multiplications that make the memory footprint large. **We use MLP and observe a very small loss in performance but significantly improving the training speed (40 min to 12 min every epoch)**, suggesting that correlations can be learned at a small cost despite large differences in features across regions.
>
> ---
>
> > **Q1**: Projection of HRV and VRV does not change its essence and still suffers from this problem.
>
> **A**: Yes, we emphasize that no single projection can avoid the problem. Therefore our superiority essentially comes from the effective combination of multiple views.
>
> ---
>
> > **Q2**: how to solve the interpolation problem for blank areas, occlusions, or object edges?
>
> **A**: It's a great question. In fact, the interpolation methods was originally designed to alleviate problems in those special areas [2]. We can discuss these cases separately, which demonstrate the advantages of the interpolation method:
>
> - **Empty area**: Empty area is marked as 0 in range view. If the neighborhoods of a query ray are all blank points, in the upsampling perspective, we believe that the query ray must also be a blank point. Therefore arbitrary weights can be correctly interpolated in empty areas. In contrast, non-interpolating methods regress distances directly from features, often failing to regress 0 values unambiguously, which results in greater noise.
> - **Occlusion**: In fact, the goal of upsampling is to add points to the observed surface, rather than considering the entire scene to be recovered. This is also the difference between upsampling and completion. Recovering occluded regions often requires strong prior information, closer to generative tasks. So occlusion is not a problem we need to solve, our goal is to generate high-resolution point clouds that respect low-resolution input.
> - **Object edges**: Explicit methods based on regression have difficulty handling sharp edges, resulting in more noise, while methods based on interpolation can more easily avoid this.
>
> ---
>
> > **Q3**: The argument for predicting the best point through feature differences lacks a theoretical basis or a more in-depth explanation.
>
> **A**: In the selection module part, we believe that conventional feature fusion methods are all feasible, such as addition, connection and attention. We use subtraction simply because 1) the $g$ we want to predict actually measures the difference between the two results, and 2) subtraction does not require any parameters. Other fusion methods can also be applied here, but this is not our focus.
>
> ---

---

> > ### Author Response · Authors · 2024-11-25
> >
> > > **Q4a** : In the selection module, the ground truth is modeled using a probabilistic approach. Is it possible to lead to a random distribution?
> >
> > **A**: Our probabilistic modeling is only intended to define the complementarity between two outcomes, i.e., the probability that one branch is better. However, our output is actually deterministic without leading to a random distribution.
> >
> > ---
> >
> > > **Q4b**: What do the final distributions and proportions from the HRV and VRV look like? It's better to provide more visualization.
> >
> > **A**: We present the distrbutions of HRV and VRV in our appendix. Intuitively, the choice of view is clearly related to the geometry. It contains some noise because both branches have been optimized.
> >
> > ---
> >
> > > **Q4c**: As opposed to this discrete selection, why not consider other fusion ways like weighted averaging or learnable fusion?
> >
> > **A**: The comparision of weighted averaging and our selection can be found in the answer to **W2**.
> >
> > ---
> >
> > > **Q5**: Other minor issues: actually, the number of points in the LiDAR point cloud is not equal to n = H × W points. Typo of $R_l$ .
> >
> > **A**: Thank you for pointing out the typos in our manuscript. We have corrected them in the updated paper.
> >
> > ---
> >
> > [1] Yang B, Pfreundschuh P, Siegwart R, et al. TULIP: Transformer for Upsampling of LiDAR Point Clouds[C]//Proceedings of the IEEE/CVF Conference on Computer Vision and Pattern Recognition. 2024: 15354-15364.
> >
> > [2] Kwon Y, Sung M, Yoon S E. Implicit LiDAR network: LiDAR super-resolution via interpolation weight prediction[C]//2022 International Conference on Robotics and Automation (ICRA). IEEE, 2022: 8424-8430.

---

> > > ### Comment · Reviewer_Qv6x · 2024-12-02
> > >
> > > I appreciate the author's efforts during the rebuttal, which alleviated my concerns. Taking into account the contribution of the paper and the comments of the other reviewers, I still maintain my score.

---

### Official Review · Reviewer_Xvya · 2024-11-08

**Soundness:** 2
**Presentation:** 3
**Contribution:** 2
**Rating:** 5
**Confidence:** 4

**Summary:**

This paper proposes a new method for LiDAR upsampling. Specifically, it first decouples the range view into two view representations: the Horizon Range View and the Vertical Range View. In each range view, the method predicts interpolation weights to interpolate the point cloud. Finally, a Contrast Selection Module is introduced to select the final result from the interpolation outputs of the two range views.

**Strengths:**

1. This paper is well-written, making it clear and easy to understand.
2. Based on the experimental results provided (though in my opinion, the experiments are not thorough enough), the method outperforms all the baselines listed in the paper.

**Weaknesses:**

1. The paper mentions that decomposing the Range View into the Horizon Range View and Vertical Range View can reduce shape distortion (L510-513). However, for LiDAR upsampling, the most intuitive approach would be to retain the original 3D representation of the point cloud, then use a backbone like Point Transformer [1] to extract per-point features. Later, for each query point, nearby points could be identified, and interpolation weights could be predicted based on the relative spatial relationships between the query point and its neighbors in 3D space. This method avoids any errors introduced by projection, so why did the authors not adopt this approach?

2. In Section 2.1 on OBJECT-LEVEL POINT CLOUD UPSAMPLING, many recent methods [2-7] are missing. Additionally, [6] can be used for point cloud sampling on the KITTI dataset, so the paper should also compare its method to [6]. Fundamentally, I believe that these methods [2-7] could be applied to large-scale LiDAR point clouds. The paper claims these methods entail "huge computational burdens," but this statement lacks experimental evidence.

3. In Equation 5, the final prediction is selected between HRV and VRV, but HRV only contains the x and y components of the 3D coordinates, while VRV contains the z component, meaning each is incomplete. Wouldn’t a more reasonable approach be to predict weights for HRV and VRV (e.g., using features F_d  and F_z as inputs to an MLP to predict weights) and then apply a weighted sum to obtain the final prediction? Why did the authors not choose this approach?

4. The paper uses depth completion as a downstream task, first downsampling the KITTI point cloud data and then upsampling with different methods. However, the upsampled point clouds here match the original point cloud resolution (which still remains sparse), which I find quite unreasonable. I suggest that object detection on point clouds should be used as the downstream task instead, as object detection performance is more sensitive to point cloud resolution. Additionally, the authors should compare object detection performance using the original KITTI point cloud data and the upsampled point clouds obtained with different methods, which would more effectively demonstrate the value of LiDAR upsampling.

5. Current point cloud generation models based on diffusion models also have potential for upsampling tasks. For example, [8] can achieve text-to-LiDAR generation. If a low-resolution range image is used as input to [8] with the text prompt "upsample 4x," how would the results from [8] compare to the method proposed in this paper?

6. This paper lacks the comparison with the baseline [9].

[1] Wu X, Jiang L, Wang P S, et al. Point Transformer V3: Simpler Faster Stronger[C]//Proceedings of the IEEE/CVF Conference on Computer Vision and Pattern Recognition. 2024: 4840-4851.

[2] Zhao W, Liu X, Zhong Z, et al. Self-supervised arbitrary-scale point clouds upsampling via implicit neural representation[C]//Proceedings of the IEEE/CVF Conference on Computer Vision and Pattern Recognition. 2022: 1999-2007.

[3] Feng W, Li J, Cai H, et al. Neural points: Point cloud representation with neural fields for arbitrary upsampling[C]//Proceedings of the IEEE/CVF Conference on Computer Vision and Pattern Recognition. 2022: 18633-18642.

[4] Qiu S, Anwar S, Barnes N. Pu-transformer: Point cloud upsampling transformer[C]//Proceedings of the Asian conference on computer vision. 2022: 2475-2493.

[5] He Y, Tang D, Zhang Y, et al. Grad-pu: Arbitrary-scale point cloud upsampling via gradient descent with learned distance functions[C]//Proceedings of the IEEE/CVF Conference on Computer Vision and Pattern Recognition. 2023: 5354-5363.

[6] Qu W, Shao Y, Meng L, et al. A Conditional Denoising Diffusion Probabilistic Model for Point Cloud Upsampling[C]//Proceedings of the IEEE/CVF Conference on Computer Vision and Pattern Recognition. 2024: 20786-20795.

[7] Rong Y, Zhou H, Xia K, et al. RepKPU: Point Cloud Upsampling with Kernel Point Representation and Deformation[C]//Proceedings of the IEEE/CVF Conference on Computer Vision and Pattern Recognition. 2024: 21050-21060.

[8] Ran H, Guizilini V, Wang Y. Towards Realistic Scene Generation with LiDAR Diffusion Models[C]//Proceedings of the IEEE/CVF Conference on Computer Vision and Pattern Recognition. 2024: 14738-14748.

[9] Helgesen S E M, Nakashima K, Tørresen J, et al. Fast LiDAR Upsampling using Conditional Diffusion Models[J]. arXiv preprint arXiv:2405.04889, 2024.

**Questions:**

1. The KITTI dataset lacks pairs of low-resolution and high-resolution point clouds for quantitatively evaluating point cloud upsampling methods. So how were the results in Table 1 generated?

2. In Equation 2, does the choice of the number of neighbors affect performance?

---

> ### Author Response · Authors · 2024-11-25
>
> We sincerely thank Reviewer `Xvya` for the valuable comments provided during this review. The point-to-point responses are as follows.
>
> ---
>
> > **W1**: Why not extract features and predict interpolation weights in 3D space?
>
> **A**: We emphasize that using range view is aiming to **maintain the line characteristics of LiDAR**. Previous work [1] has proven through complete comparative experiments that point-based methods cannot restore the scan line structure of LiDAR, and are significantly behind range view-based methods in various indicators. In order to further illustrate this problem, We compare the performance of the PUDM [2] and our method on the CARLA dataset (4 $\times$ upsampling) in Table 1. The experimental results show that it is much slower and has lower accuracy.
>
> ---
>
> > **W2**: Missing citations to recent object-level methods. Lack of camparison to object-based point cloud upsampling methods.
>
> **A**: Thanks for your reminder, we have added citations to these works in the updated paper. However, due to the scanning characteristics of LiDAR, the acquired point clouds are larges-cale, uneven in density, and have line characteristics, which is very different from object-level point clouds. Many previous works [1, 3, 5-6] stated that range view-based methods are far superior to point-based methods, both in efficiency and accuracy. We briefly compared the recent object-level method PUDM [2] , and the results show that **our method far outperforms [2] because [2] cannot be adapted to LiDAR data**. The specific results are shown in the table below.
>
> **Table1**: Comparison of object-level method [2] and our WIN.
>
> |  Methods  |   MAE(m)   |    IoU     | Time(sec) |
> | :-------: | :--------: | :--------: | :-------: |
> |   PUDM    |  12.2957   |   0.0713   |   4.75    |
> | WIN(Ours) | **1.3834** | **0.4561** | **0.03**  |
>
> ---
>
>
> > **W3**: Why not apply a weighted sum to obtain the final prediction?
>
> **A**: The motivation for using selection rather than weighted sum is twofold: 1) the two views have a clear geometric meaning, while the weighted sum does not, and 2) the weighted sum prevents the model from fully optimizing the two branches. We added a comparison experiment with the weighted sum method and the results show that our selection method is more advantageous as shown in the Table 2. Poor performance on IoU suggests that weighted sum methods may lose geometric information. We also added the visulization of the complementary of HRV and VRV in A.1 Fig
>
> **Table2**: Comparison of weighted sum and selection.
>
> |     Methods     |   MAE(m)   |    IoU     |
> | :-------------: | :--------: | :--------: |
> |       ILN       |   1.4168   |   0.3927   |
> |  Weighted Sum   |   1.4131   |   0.3604   |
> | Selection(Ours) | **1.3834** | **0.4561** |
>
> ---
>
> > **W4**: the upsampled point clouds matching the original point cloud resolution is unreasonable.
>
> **A**: Since the KITTI data does not have a 128-line ground-truth, we can only provide a quantitative evaluation by downsampling and upsampling. Similarly, we can only employ this setup on downstream tasks to evaluate different models, as the current methods can not generalize to every input and output resolutions. We argue that our experiments are still meaningful, because **improving the availability of sparse point clouds could facilitate the use of lightweight, cost-effective sensors in practical applications [4, 7, 10]**.
>
> ---
>
> > **W5**: The authors should compare object detection performance using the original KITTI point cloud data and the upsampled point clouds obtained with different methods.
>
> **A**: As mentioned previously, we cannot apply all models to the original KITTI data. So we still upsample the 16-line point cloud into a 64-line point cloud and use the pre-trained model to provide object detection results. The results in the Table 3 show that our **WIN can effectively improve the object detection result by upsample the low-resolution point clouds.**
>
> **Table3**: Results of object detection performance by different upsampling methods.
>
> |  Method   |   Easy    | Moderate  |   Hard    |
> | :-------: | :-------: | :-------: | :-------: |
> |  16-line  |   10.05   |   9.03    |    8.8    |
> |    ILN    |   38.29   |   28.61   |   23.67   |
> | WIN(Ours) | **40.11** | **28.80** | **26.67** |
>
> ---
>
>
> > **W6**:  [12] can achieve text-to-LiDAR generation. If a low-resolution range image is used as input to [12] with the text prompt "upsample 4x," how would the results from [12] compare to the method proposed in this paper?
>
> **A**: Lidar upsampling task typically serve low-cost chips and real-time responsive autonomous driving systems that require models that are easy to deploy. Diffusion-based upsampling methods are all inadequate for this task because the sampling is too slow.
>
> ---

---

> > ### Author Response · Authors · 2024-11-25
> >
> > > **W7**: This paper lacks the comparison with the baseline "Fast LiDAR Upsampling using Conditional Diffusion Models".
> >
> > **A**: We have followed this work, but as mentioned before, diffusion-based methods are not suitable for our task. In addition, we don't find any open source code.
> >
> > ---
> > > **Q1**:The KITTI dataset lacks low- and high-resolution point cloud pairs for quantitatively evaluating point cloud up-sampling methods. So how were the results in Table 1 generated?
> >
> > **A**: We followed the setup of the previous methods[1, 3-7, 10-11] by downsampling the 64-line point cloud to 16 lines and using the 64-line point cloud as ground truth.
> >
> > ---
> >
> > > **Q2** :In Eq. 2, does the choice of the number of neighbors affect the performance?
> >
> > **A**: Thank you for your constructive comments. This issue has been studied in ILN [10] and is not the focus of our solution design. Thus, for fair comparison with interpolation-based methods [10], we followed ILN [10] to set the number of neighbors to 4. Experimental result show that it is enough for the task.
> >
> > ---
> >
> > [1] Eskandar G, Sudarsan S, Guirguis K, et al. Hals: A height-aware lidar super-resolution framework for autonomous driving[J]. arXiv preprint arXiv:2202.03901, 2022.
> >
> > [2] Qu W, Shao Y, Meng L, et al. A Conditional Denoising Diffusion Probabilistic Model for Point Cloud Upsampling[C]//Proceedings of the IEEE/CVF Conference on Computer Vision and Pattern Recognition. 2024: 20786-20795.
> >
> > [3] Jung Y, Seo S W, Kim S W. Fast point clouds upsampling with uncertainty quantification for autonomous vehicles[C]//2022 International Conference on Robotics and Automation (ICRA). IEEE, 2022: 7776-7782.
> >
> > [4] Tian D, Zhao D, Cheng D, et al. Lidar super-resolution based on segmentation and geometric analysis[J]. IEEE Transactions on Instrumentation and Measurement, 2022, 71: 1-17.
> >
> > [5] Savkin A, Wang Y, Wirkert S, et al. Lidar upsampling with sliced wasserstein distance[J]. IEEE Robotics and Automation Letters, 2022, 8(1): 392-399.
> >
> > [6] Chen T Y, Hsiao C C, Huang C C. Density-imbalance-eased lidar point cloud upsampling via feature consistency learning[J]. IEEE Transactions on Intelligent Vehicles, 2022, 8(4): 2875-2887.
> >
> > [7] Chen C, Jin A, Wang Z, et al. SGSR-Net: Structure Semantics Guided LiDAR Super-Resolution Network for Indoor LiDAR SLAM[J]. IEEE Transactions on Multimedia, 2023, 26: 1842-1854.
> >
> > [8] Nakashima K, Kurazume R. Lidar data synthesis with denoising diffusion probabilistic models[C]//2024 IEEE International Conference on Robotics and Automation (ICRA). IEEE, 2024: 14724-14731.
> >
> > [9] Zyrianov V, Zhu X, Wang S. Learning to generate realistic lidar point clouds[C]//European Conference on Computer Vision. Cham: Springer Nature Switzerland, 2022: 17-35.
> >
> > [10] Kwon Y, Sung M, Yoon S E. Implicit LiDAR network: LiDAR super-resolution via interpolation weight prediction[C]//2022 International Conference on Robotics and Automation (ICRA). IEEE, 2022: 8424-8430.
> >
> > [11] Yang B, Pfreundschuh P, Siegwart R, et al. TULIP: Transformer for Upsampling of LiDAR Point Clouds[C]//Proceedings of the IEEE/CVF Conference on Computer Vision and Pattern Recognition. 2024: 15354-15364.
> >
> > [12] Ran H, Guizilini V, Wang Y. Towards Realistic Scene Generation with LiDAR Diffusion Models[C]//Proceedings of the IEEE/CVF Conference on Computer Vision and Pattern Recognition. 2024: 14738-14748.

---

### Comment · Area_Chair_AnQB · 2024-11-27
**Reminder: Last day for author feedback**

This is a reminder that today is the last day allotted for author feedback. If there are any more last minute comments, please send them by today.

---

### Meta-Review · Area_Chair_AnQB · 2024-12-19

**Metareview:**

The authors proposed a method for LiDAR upsampling where they consider a horizon and vertical range view. They propose to predict interpolation weights for each range view for interpolating the point cloud, which is followed by a selection mechanism. We have read the referee reports and the author responses. The referees raised several critical points in the choice of representation, its novelty (bares similarity to existing work [A, B, C]), and experiments (missing comparisons with baselines). There were also concerns raised in terms of the method being trivial or not complex (which may stem from possible overclaim in the introduction); we do not believe that a complex method is necessarily ``better'' or has ``more contribution/novelty'' so long as it is algorithmically sound and should be reflected in the writing. While the authors have provided additional experiments, the referees feel that the responses were unable to fully address their concerns. We believe that additional experiments provided by the authors do improve the quality of the manuscript and encourage the authors to incorporate the points discussed into their next revision.

[A] Ku et al. Joint 3D Proposal Generation and Object Detection from View Aggregation. IROS 2018.

[B] Chen et al. Multi-View 3D Object Detection Network for Autonomous Driving. CVPR 2017.

[C] Liang et al. Deep Continuous Fusion for Multi-Sensor 3D Object Detection. ECCV 2018.

**Additional Comments On Reviewer Discussion:**

The referees raised several critical points in the choice of representation, its novelty (bares similarity to existing work [A, B, C]), and experiments (missing comparisons with baselines). There were also concerns raised in terms of the method being trivial or not complex (which may stem from possible overclaim in the introduction); we do not believe that a more complexity method implies more contribution/novelty, so long as it is algorithmically sound and should be reflected in the writing.

---

### Decision · Program_Chairs · 2025-01-22

Reject